# Dynamic cybergenetic control of bacterial co-culture composition via optogenetic feedback

Joaquín Gutiérrez Mena [1,2], Sant Kumar [1,2] & Mustafa Khammash [1]✉

Communities of microbes play important roles in natural environments and hold great potential for deploying division-of-labor strategies in synthetic biology and bioproduction. However, the difficulty of controlling the composition of microbial consortia over time hinders their optimal use in many applications. Here, we present a fully automated, high-throughput platform that combines real-time measurements and computer-controlled optogenetic modulation of bacterial growth to implement precise and robust compositional control of a two-strain *E. coli* community. In addition, we develop a general framework for dynamic modeling of synthetic genetic circuits in the physiological context of *E. coli* and use a host-aware model to determine the optimal control parameters of our closed-loop compositional control system. Our platform succeeds in stabilizing the strain ratio of multiple parallel co-cultures at arbitrary levels and in changing these targets over time, opening the door for the implementation of dynamic compositional programs in synthetic bacterial communities.

Despite their microscopic size, bacteria constitute the second largest segment of the total biomass of our planet[1], having a direct impact on a huge range of processes: from biogeochemical cycles to human health. The ability of microbes to perform such vastly diverse tasks stems, to a large extent, from their existence in complex communities of interacting, specialized individuals[2]. Bacterial species can behave differently within such a community than in isolation and there is a growing interest in probing how interspecies interactions give rise to the overall properties of the community[3–5] in terms of its joint metabolism[6,7], spatial arrangement[8,9] and dynamic composition[10,11]. The importance of these studies will only grow in the future, as researchers try to disentangle the complex interplay between microbial communities and their environment.

Engineered microbial communities also offer an enticing prospect for applications in the field of synthetic biology[12], distributed computing[13–16], multicellular control[17], and bioproduction[18,19]. By exploiting the natural capacities of member species, diverse communities can carry out tasks that would be out of reach for monoclonal populations[20]. In addition, in a division-of-labor approach, large genetic

circuits and pathways can be broken up into subsets that are placed inside different member species, offering a natural compartmentalization that improves modularity and allows for the reuse of parts in different contexts[21,22]. More importantly, the resulting community can carry out its target function more efficiently than an isogenic, engineered population, because individual cells bear only a fraction of the total production burden[17,23–25]. As applications of synthetic consortia continue to emerge, it becomes increasingly essential to develop methods for controlling such communities in time and space[26,27].

One of the biggest challenges for the widespread deployment of applications that rely on microbial consortia stems from the *competitive exclusion principle*, which states that, in the absence of stabilizing interactions, a community of species competing over the same ecological niche will be overtaken by its fastest-growing member[28]. Since members of synthetic consortia generally compete for space and shared nutrients, this implies that it is difficult to maintain a stable community composition over time. Moreover, the problem is exacerbated by differences in growth rate that arise when the members of the consortium carry different loads from exogenous genetic

[1]Department of Biosystems Science and Engineering (D-BSSE), ETH Zürich, Mattenstrasse 26, 4058 Basel, Switzerland. [2]These authors contributed equally: Joaquín Gutiérrez Mena, Sant Kumar. ✉e-mail: mustafa.khammash@bsse.ethz.ch

components. Methods for stabilizing co-existence have largely focused on self-limiting populations[29,30] and on the introduction of stabilizing interactions between members of the consortium[6,21,31–35]. Although these approaches succeed in producing stable community dynamics, both the dynamic behavior and the attainable strain ratios at equilibrium are often hard-wired properties of the system. Moreover, genetically engineering strains that interact with each other and optimizing the strength of those interactions can be a long and complex process. Therefore, these methods do not offer a general solution that can be easily adapted to different contexts and applications.

This constitutes a serious bottleneck, because the performance of microbial co-cultures at biocomputing, biosensing and bioproduction tasks depends crucially on the maintenance of optimal population ratios and more complex applications might even require the ability to change those ratios over time. Most applications to date rely on strategies that lack robustness, such as manual tuning of initial inoculation ratios[22,36,37] or open-loop control[38]. Thus, achieving precise, robust, and flexible control over the dynamic composition of a co-culture would add a powerful tuning knob for the optimization of future applications.

In silico feedback controllers offer an attractive alternative to embedded (genetically engineered) controllers[39]. In this approach, a biological system is interfaced with a controller algorithm that is executed externally by a computer[40,41], facilitating the implementation of arbitrary controller architectures and providing flexible control over the properties of the closed-loop system[42]. Although engineered cells still require components that allow them to react to the inputs from the controller, they are freed from the genetic load of the controller itself[43], releasing cellular resources that can be allotted to the expression of application-specific products or circuits. In spite of some recent attempts at controlling microbial communities by interfacing them with a computer[44–46], the full potential of in silico feedback for this task has yet remained untapped.

Here, we present a strategy that integrates optogenetic control of cellular growth and in silico feedback to maintain arbitrary strain ratios in an otherwise unstable bacterial co-culture (Fig. 1a). For this, we combine two engineered *E. coli* strains: a constitutive strain that grows at a fixed rate and a photophilic strain, whose growth rate can be modulated by external light inputs. In this system, we can effectively steer the composition of the two-strain consortium using light (Fig. 1b). We further develop a custom-built, generic and modular framework for automated sampling, which we integrate with a commercially available culturing system that we modified for optogenetic applications. The result is a fully automated platform for high-throughput continuous culturing, sampling and light stimulation, which allows us to monitor the composition of multiple co-cultures in parallel with high time resolution using flow cytometry. By implementing a suitable proportional-integral-derivative (PID) controller that acts on the strain ratio, we demonstrate in silico closed-loop control of co-culture composition using this automated platform (Fig. 1c). In addition, we develop a framework for host-aware mathematical modeling of synthetic genetic circuits. The framework takes into account the deleterious effects of gene-expression burden on host growth and the

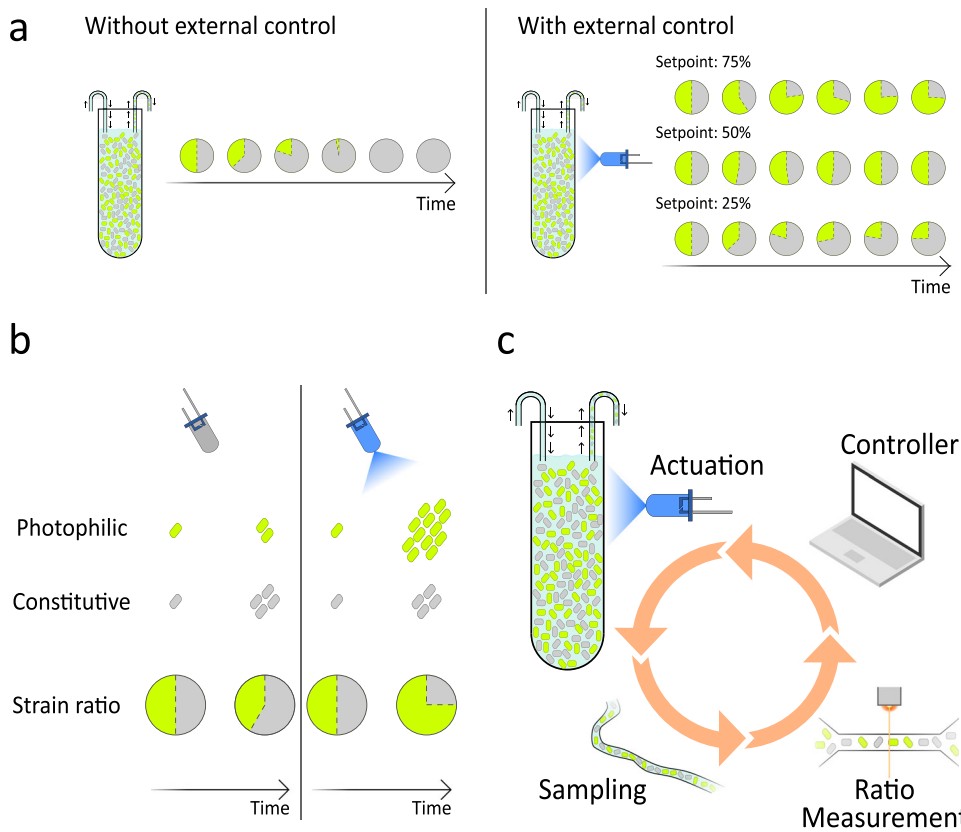

**Fig. 1 | Compositional control of a bacterial co-culture via optogenetic feedback. a** Because of the competitive exclusion principle, two non-interacting strains that compete for common space and resources cannot stably coexist in a co-culture (left). In this study, we show that the precise composition of such a co-culture can be modulated and stabilized at arbitrary strain ratios using external optogenetic feedback (right). **b** Our co-cultures contain a constitutive strain, which grows at a fixed rate independent of light, and a photophilic strain, whose growth is stimulated by blue-light. The fate of the co-culture can be controlled through the choice of external light inputs, with the constitutive strain taking over the culture in the dark and the photophilic strain taking over under strong illumination. **c** We implement external optogenetic feedback in a fully automated platform that includes a continuous culture with an LED for the delivery of light inputs, automated sampling coupled to a flow cytometer to allow us to monitor the strain ratio with high temporal resolution and a controller algorithm, running on a computer, that updates the intensity of the input light based on the current state of the co-culture and the control objective.

consequences of resource limitation for circuit performance, while reproducing the empirical relations between growth rate and cellular resource pools known as bacterial growth laws[47]. Our framework seamlessly extends existing ODE models of arbitrary circuits to simulate the context of an *E. coli* host, without adding any free parameters to the original model. We apply the framework to our photophilic strain and show that it accurately captures the dynamics of the optogenetic growth-control circuit. This allows us to simulate the behavior of the closed-loop co-culture system and to determine the optimal parameters for our PID controller through a computational screening procedure. Using our automated platform with the optimized PID controller, we demonstrate that we can force the photophilic-constitutive consortium to maintain arbitrary strain ratios for over 80 bacterial generations and that the strain ratio can track dynamically changing setpoints. The possibility of implementing arbitrary, dynamic profiles in the composition of bacterial communities through a strategy that is orthogonal to intercellular communication channels offers exciting opportunities, both for studies of microbial ecology and for the optimization of synthetic biology and bioproduction applications.

## Results

### Optogenetic control of antibiotic resistance enables fast modulation of bacterial growth

In order to control the relative abundances of strains in a co-culture, we needed to grow the cells in an environment that allows us to have precise control over their growth rates. However, we wanted to avoid growth-control approaches that rely on auxotrophy and targeted expression of bottleneck metabolic enzymes[41] to ensure that our platform can be used in the broadest possible contexts. Therefore, we decided to grow our strains in the presence of sub-lethal concentrations of a bacteriostatic antibiotic, which makes their rate of growth dependent on the expression levels of a resistance-conferring enzyme.

For this purpose, we use chloramphenicol, a bacteriostatic antibiotic that inhibits growth by binding to the ribosomal 50S subunit and preventing peptidyl transfer[48]. In the absence of resistance, sub-saturating concentrations of chloramphenicol lead to a dose-dependent reduction in growth rate[49]. The enzyme chloramphenicol acetyltransferase (CAT) catalyzes the conversion of chloramphenicol to a non-toxic product and thereby renders CAT-producing cells resistant to the antibiotic[50]. We reasoned that, if we exposed the cells to a fixed external concentration of chloramphenicol, then controlling the expression of CAT would allow us to modulate the growth rate of the cell, because low levels of the enzyme would alleviate the effect of the antibiotic without abolishing it entirely.

We begin by building a strain of *E. coli* whose growth rate can be modulated through blue-light illumination. For this, we use an optogenetic tool developed in our lab, the opto-T7 polymerase[51], to express CAT in a dose-dependent manner (Fig. 2a). Opto-T7 consists of a polymerase from T7 bacteriophage which is split into two catalytically inert parts that are fused to matching light-inducible heterodimerization domains[51]. When exposed to blue light, complementary parts interact in a dose-dependent manner and reconstitute a functional polymerase unit, which can then initiate transcription of the CAT gene from a T7-recognized promoter. In our design, we express the components of the opto-T7 from a low-copy plasmid (pSC101) and place the target gene in a separate plasmid (p15A). We also fused an mCherry fluorescent protein to the C-terminus of CAT to have a direct readout of the resistance expression levels. Finally, our light-responsive strain carries a constitutive mVenus expression cassette in the chromosome to be able to tell it apart from the other non-fluorescent strain in a co-culture. Since these cells grow more rapidly in the presence of light, we refer to them as the photophilic strain.

We chose to work with an opto-T7 polymerase because it offers many advantages compared to other available tools[52]. On the one hand, it is a one-component system that requires no co-factors,

imposing a lower expression burden on cells. Its orthogonality with respect to the endogenous transcriptional machinery reduces the chances of undesired interactions with cellular processes, as well as with other synthetic circuits that might be placed in the cell alongside the growth control module in future applications. Most importantly, it supports a high fold change in expression between dark and illuminated conditions and both its ON-switching and OFF-switching dynamics are fast.

In agreement with these properties, the growth rate of the photophilic strain displays a well-tunable, gradual response to blue-light intensity (Fig. 2b and Supplementary Fig. 1). When grown in the presence of 10.5 μM chloramphenicol, we can precisely set the growth rate of the strain in a range between ~1 h$^{-1}$ and ~1.9 h$^{-1}$ by modulating the intensity of input light. Furthermore, we can also vary the external antibiotic concentration to obtain different fold changes in growth rate between a dark environment and maximum illumination (Supplementary Fig. 2).

The use of an antibiotic-inactivating enzyme, such as CAT, for dynamic control of the growth rate, requires that the cells are grown in a continuous culture. The reason is that, if the antibiotic is not continuously replenished, the action of the resistance enzyme causes the external concentration of antibiotic to drop over the course of the experiment. As a culturing platform, we use eVOLVER, a commercially available solution for monitoring several turbidostat cultures in parallel[53]. Automated, high-frequency measurements of the optical density allowed us to monitor the growth rate dynamics of the cultures with high resolution.

Notably, the dynamics of the photophilic strain are fast (Fig. 2c) with cells reaching a new steady-state around two hours after an increase in light intensity (upshift) and four hours after a decrease in light intensity (downshift). We attribute the asymmetry in the dynamics to the fact that the resistance enzyme is not tagged for degradation and so its removal relies solely on dilution through cell growth, whereas the dynamics of upshifts are dictated by the timescales of the gene expression process. In both up- and downshifts, we measured a delay of around one hour before there was an appreciable change in growth rate.

Finally, we built a set of constitutive strains in which the expression levels of CAT are fixed and independent of light (Fig. 2a). For this, we built several expression cassettes with different promoters and ribosome-binding sites to achieve different expression levels. In the presence of our chosen concentration of chloramphenicol, these strains indeed exhibit different growth rates and are unaffected by light.

The growth rate of four of these strains is plotted in Fig. 2d, along with the maximal and minimal growth rates of the photophilic strain for comparison. The growth rate of the constitutive strain should lie between these extremes, so that an external controller can both increase and decrease the relative abundance of the photophilic strain in the co-culture by delivering the appropriate intensity of light. Three of the constructed strains met this requirement. We selected one of these, bJAG235, for future co-culture experiments with the photophilic strain. In the remainder of the paper, we refer to bJAG235 simply as the constitutive strain.

### *evotron*—an automated high-throughput culture, sampling, and light-stimulation platform

In this study, we needed a continuous (possibly high-throughput) cell culture platform integrated with an automated sampling setup for fetching culture samples periodically to a measurement device, and an integrated light-delivery device to implement a fully automated closed-loop feedback control over the bacterial co-culture composition. To achieve this objective, we developed a generic and modular platform —*evotron*—, as illustrated in Fig. 3.

For continuous cell culture, we considered the eVOLVER framework[53], which provides an integrated setup for high-throughput

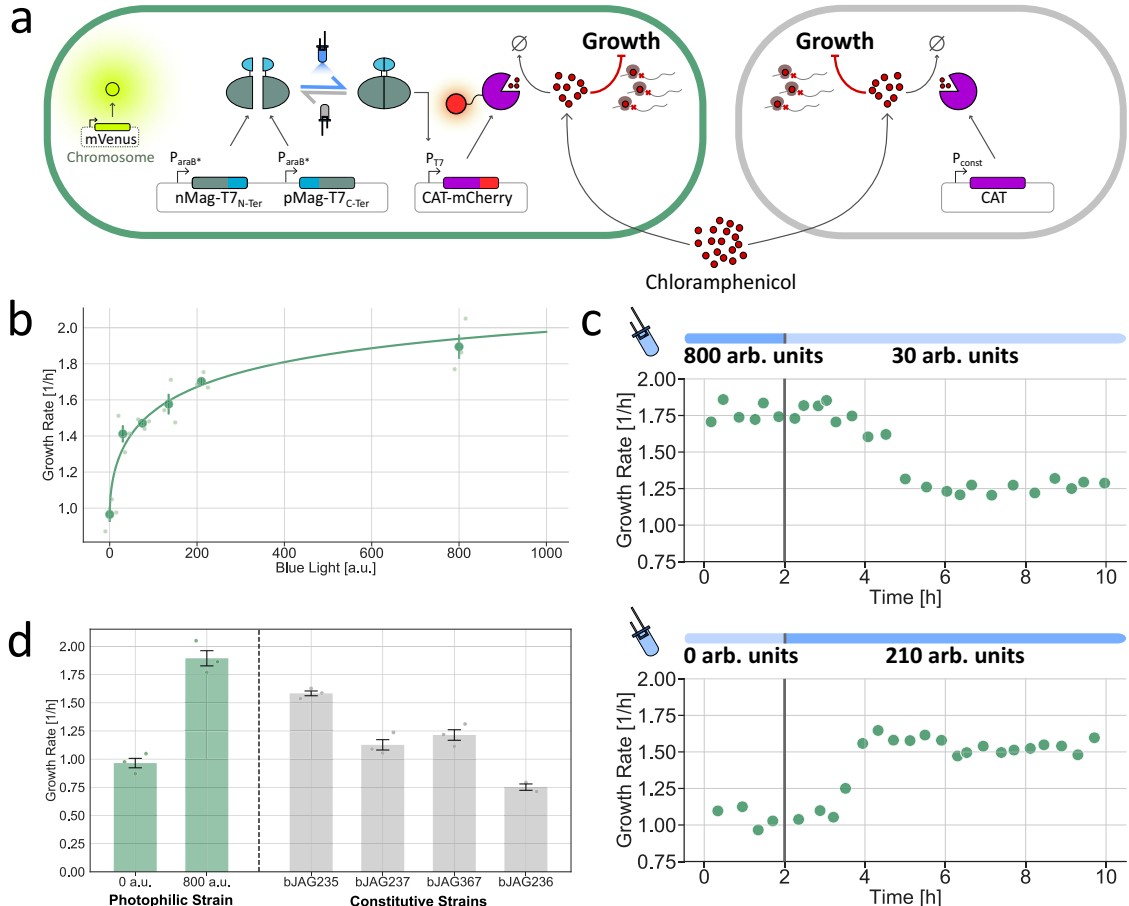

**Fig. 2 | Optogenetic control of cellular growth rate. a** Schematic of the genetic circuits used in this study. Control over the growth rate is achieved by growing *E. coli* in the presence of a fixed concentration of chloramphenicol and titrating the expression of chloramphenicol acetyltransferase (CAT), an enzyme that inactivates the antibiotic. The photophilic strain (top) carries a split T7 polymerase fused to light-inducible heterodimerization domains and the CAT gene, placed under control of a T7-promoter. Blue-light illumination leads to reconstitution of active T7-polymerase units and production of CAT, which in turn results in faster growth. In the constitutive strain (bottom), the CAT gene is expressed from a constitutive promoter, so that the growth rate is independent of light. **b** Dose-response of the growth rate of the photophilic strain to blue-light intensity in the presence of sub-lethal concentrations of chloramphenicol (10.5 μM). Data are presented as mean values +/− SEM, with the median steady-state growth rate of individual biological replicates shown as data points with different transparency (*n* = 3 biologically

independent samples). The raw time-course data used to determine the steady-state growth is presented in Supplementary Fig. 1. **c** Dynamic response of the photophilic strain to a step changes in blue-light intensity. The transient phases of growth downshifts and upshifts have different duration. **d** Comparison between the controllable range of growth rates of the photophilic strain (minimum and maximum) and the growth rates of several constitutive strains. Data are presented as mean values +/− SEM, with the median steady-state growth rate of individual biological replicates shown as data points (*n* = 3 biologically independent samples). For the composition of a photophilic-constitutive co-culture to be controllable, the growth rate of the constitutive strain must lie in between the extremes of the photophilic strain. Therefore, strain bJAG236 cannot be used together with the photophilic strain and was only measured in duplicates (*n* = 2 biologically independent samples). Source data are provided as a Source Data file.

(16 in parallel) and automated cell culture for long-duration experiments. We re-designed their smart sleeves (tube-holders) and glass vial caps to accommodate O-rings so as to prevent wobbling of glass vials when placed inside the sleeves (Fig. 3a, left). This resulted in considerably stable and consistent OD (optical density) sensor readings (Supplementary Fig. 3), even at the low cell culture densities typically maintained during our experiments. We also integrated one blue LED per sleeve and modified the embedded firmware to allow for controlled, time-varying, and independent light-illumination of parallel cell cultures during the course of optogenetic experiments. For maintaining cell culture density within a desired range, we used the built-in turbidostat functionality (Fig. 3a, center) of the eVOLVER framework, which performs controlled dilution and culture removal steps in a feedback-controlled manner. After multiple trials, we tuned the turbidostat OD-regulation controller parameters providing a better OD setpoint tracking with less media consumption (Supplementary Fig. 4). OD of parallel cell cultures were independently maintained

between 0.1 and 0.15 in all of our experiments. The eVOLVER framework also provided a built-in function to compute growth rate in near real-time using the measured OD trajectory data fluctuating between 0.1 and 0.15 (Fig. 3a, right).

To realize closed-loop optogenetic feedback control over bacterial co-cultures, we integrated a flow-cytometry measurement device with our cell culture optogenetic platform via a generic automated sampling framework, as shown in Fig. 3b. We placed our modified eVOLVER platform on the deck of an Opentrons OT-2 Robot in such a way that all 16 of eVOLVER sleeves were laid within the accessible working region of the OT-2 pipette head. We also designed an adapter (3D printed) for the pipette head to hold a sampling needle, which can be lowered through the modified vial cap into the cell culture in individual vials for sampling. This configuration allowed us to move the sampling needle over any of the 16 eVOLVER sleeves and lower it down inside the cell culture using a custom-developed OT-2 protocol code. Bottles with three cleaning solutions ((1) sterile $H_2O$, (2) 2% bleach solution, (3) sterile $H_2O$) were

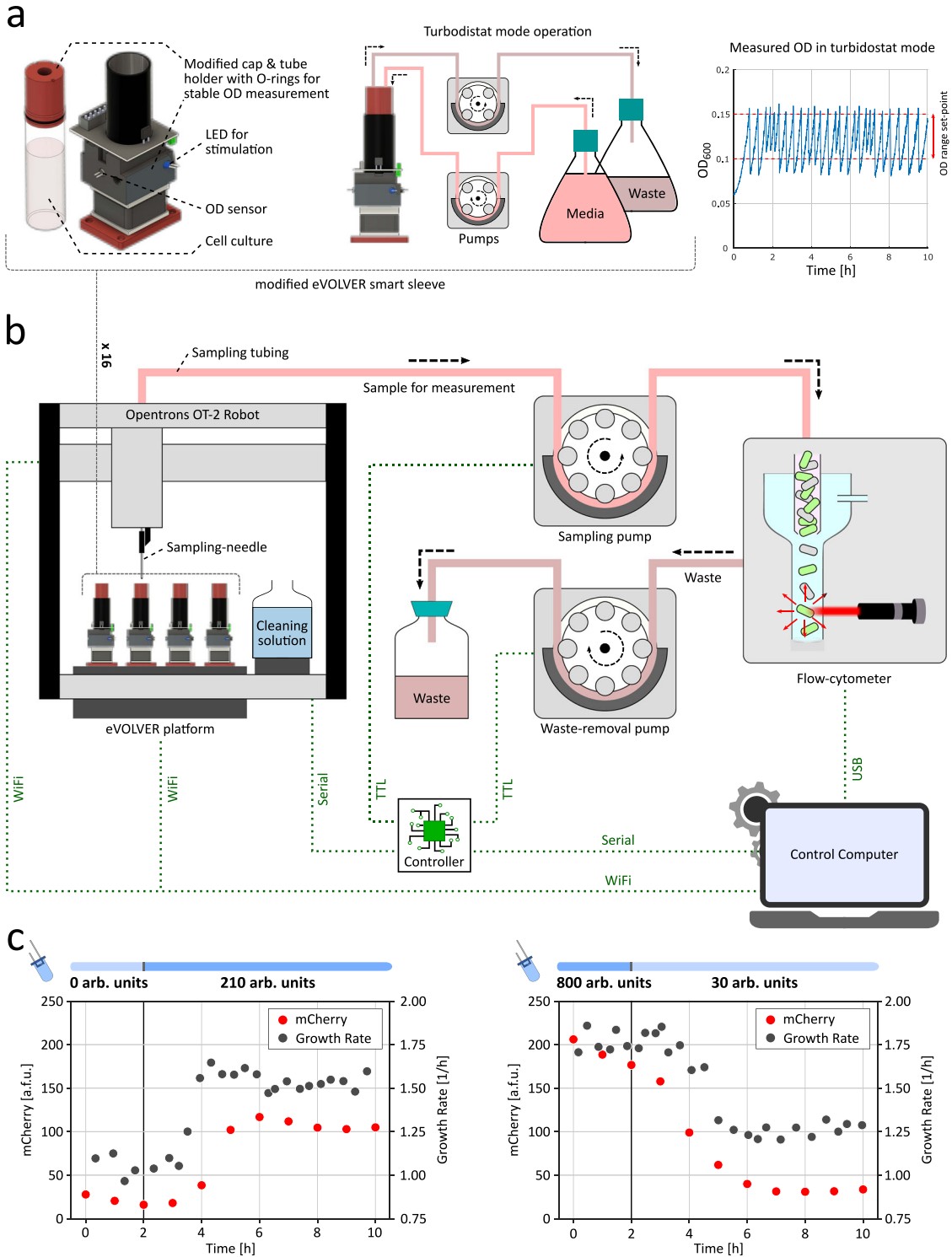

also placed on the OT-2 deck for cleaning the sampling needle and the sampling tubing after every sampling event in order to prevent cross-contamination between successive samples or different cell cultures (Supplementary Fig. 5). Furthermore, we connected the sampling needle to the flow-cytometer sample vial with a flexible silicone tubing routed via a peristaltic sampling pump for drawing samples from target cell cultures. A separate tubing connected the flow-cytometer sample vial to a waste bottle via another peristaltic pump for removing the residual sample culture from the cytometer. Together with this integrated hardware elements, we programmed the sampling and measurement operation to follow four steps: first, move and lower the

sampling needle into a desired eVOLVER vial with cell culture, and draw 0.5 ml sample to the flow-cytometer sample vial using the sampling pump; second, start cytometry measurement; third, once the measurement is finished, remove the left residual sample from the cytometer sample vial; fourth, move and lower the sampling-needle to the three sampling solutions in succession while running the sampling pump and waste pump sequentially, thus moving the cleaning solutions through the entire sampling path.

We developed a primary routine running on a control computer and a secondary sub-routine running on an Arduino controller for sequential execution of the above-mentioned steps. Provisions for

**Fig. 3 | *evotron*—automated high-throughput culture, sampling, and light-stimulation platform.** We used a modified eVOLVER platform[53] for maintaining and stimulating our target cell culture, and developed an Opentrons OT-2 Robot-based generic and modular setup to facilitate automated periodic sampling and measurement in our experiments. **a** Left: Modified eVOLVER smart sleeve. We re-designed the glass vial cap and the tube-holder for stable and consistent OD (optical density) sensor measurements (Supplementary Fig. 3). We also integrated one blue LED per sleeve in the framework for dynamic light-stimulation of the target culture during an optogenetic experiment. Center: Turbidostat-mode operation. The modified eVOLVER platform was used in turbidostat mode to maintain cell culture density within a desired range during the course of an experiment via a controlled dilution and cell-culture removal process. Right: OD measurements during an experiment. Cell density was maintained within a 0.1–0.15 OD range in all of our experiments. **b** Opentrons OT-2 Robot-based automated sampling platform. We placed the modified eVOLVER platform on the OT-2 deck, ensuring that all 16 sleeves stayed within the accessible region of the OT-2 pipette head. The pipette head was fitted with a custom-designed adapter (3D printed)

holding a sampling-needle that can be lowered into the cell culture in individual vials for sampling. We also placed cleaning solutions on the OT-2 deck to clean the sampling-needle and tubing after each sampling in order to avoid cross-contamination. At every sampling instance, the sampling-needle is moved to the desired culture vial and lowered into it. A sampling pump then extracts around 0.5 ml of cell culture into a sampling tubing, and draws it through the tubing into a flow-cytometer sample vial. Once the cytometry measurement is done, a separate waste-pump removes the left-over sample from the flow-cytometer sample vial. The sampling-needle is then moved and lowered into the cleaning solutions one-by-one, with sampling-pump and waste-pump running sequentially to clean the entire culture sample path. We developed a primary and secondary routines running on a control computer and an embedded controller respectively to execute sampling steps, run feedback-control algorithms over the measurement, and set the stimulating LED intensity accordingly. Communication channels between different elements are shown with dotted green lines in the figure. **c** Dynamic responses from Fig. 2c are shown with the automated fluorescence (mCherry) measurements obtained using *evotron* platform. Source data are provided as a Source Data file.

feedback-controller implementation were also provided in the primary routine design allowing one to implement measurement-based control algorithms (e.g., PID controller illustrated in Fig. 6a), computing light stimulation intensities which are then applied to the respective target cell cultures accordingly. The control computer and Arduino controller were connected to eVOLVER, OT-2, pumps, and flow-cytometer via different communication channels as illustrated in Fig. 3b. Using this automated platform, we were able to observe the dynamic step response of the photophilic strain (Fig. 2a) both in terms of growth rate as well as fluorescence (mCherry) intensity changes, as shown in Fig. 3c. *evotron* thus enables simultaneous, high-resolution monitoring of fluorescence and growth.

Although we used a modified eVOLVER platform for continuous cell-culture maintenance in our *evotron* framework, the generic, modular, and simple design of our automated sampling setup allows one to easily integrate any other cell culture platform, which can be placed on the large deck of the OT-2 robot. Associated control routines can be easily modified to incorporate those different platforms and integrate other measurement devices as desired. While we were working on the manuscript, a similar approach for automated measurements and reactive control operations was published[44]. Please refer to Supplementary Text Section 1 for a brief comparison.

## A framework for host-aware modeling of synthetic genetic circuits in *E. coli*

Automated sampling allows us to monitor the composition of the co-culture over time—a prerequisite for any implementation of real-time optogenetic feedback on the strain ratio. As a next step, we aimed to develop a mathematical model of the co-culture dynamics that we can use for model-guided design of an optimized control strategy. Since the dynamics of the co-culture are fully determined by those of the photophilic strain, we first derived a model that accurately captures that strain's response to light.

We reasoned that a conventional model that does not consider the physiology of the cell would not be able to capture the internal feed-back interactions between the growth-control circuit and its host. In fact, the circuit directly affects the growth rate and this, in turn, changes the dilution rate and the physiological state of the cell which, as a result, causes a feedback action on the expression of the genes that constitute the circuit. The coupling between the circuit and the host arises from three main factors. First, circuit components are diluted at a rate proportional to the cell's growth rate. Second, translation of the proteins of the circuit is carried out by the host ribosomes, which are affected by the intracellular concentration of chloramphenicol. Therefore, the inactivation of chloramphenicol by the circuit feeds back on the circuits expression levels. Finally, the host responds to the presence of the antibiotic by changing its ribosomal content, leading to a broad

reconfiguration of the protein contents of the cell. This dynamic adaptation impacts the circuit's expression levels and dynamics.

To take these aspects into account, we developed a modeling framework that captures all these internal feedbacks between the growth-control circuit and the physiological state of the cell. We further aimed to make the framework as simple and general as possible, so that it can be applied to genetic circuits beyond the particular example of the photophilic strain. For this, we turned to the type of proteome-partition models that were first put forward as interpretations of so-called bacterial growth laws—a set of empirical relations describing how the cell's ribosomal content scales with its growth rate[47].

Figure 4a summarizes these growth laws, as well as the proteome-partition model proposed by Scott et al.[47]. When the growth rate of exponentially dividing *E. coli* is reduced by diminishing the nutrient quality, the ribosomal fraction of the proteome, $\Phi_R$, shrinks proportionally. In contrast, when the reduction in growth rate is caused by exposure to translation-inhibiting antibiotics, the ribosomal content increases (For details, see ref. 47 and Supplementary Text Section 2). In proteome-partition models, these observations are interpreted as shifts in the way that the finite-sized proteome is allocated to two broad, co-regulated categories. In particular, the expansion of the ribosomal fraction necessarily comes at a cost to the catabolic fraction, $\Phi_P$, a regulatory strategy that is thought to reflect the tight balance between anabolic and catabolic fluxes in exponential growth[47,54]. The proteome-partition model quantitatively relates the relative sizes of $\Phi_R$ and $\Phi_P$ to the growth rate, as a function of only two parameters that describe the nutrient quality and potential effect of translation-inhibiting factors.

Our framework consists of an adaptation of the original proteome-partition model to incorporate the dynamics of an arbitrary network of $u$ exogenous genes, $x_1, \cdots, x_u$, whose expression both occupies a fraction of the cell's limited proteome and relies on the host's transcriptional and translational machinery (Fig. 4b). For this, we include a further fraction of the proteome, the synthetic fraction $\Phi_S$, which is composed of the proteins of the circuit, $X_i$. We model the time evolution of the concentrations of these proteins through a system of ordinary differential equations (ODEs) and then use a simple conversion rule to relate the concentration of exogenous proteins to the synthetic proteome fraction

$$\Phi_S = \frac{\sum_{i=1}^{u} n_{X_i} X_i}{\rho_{\text{cell}}} \qquad (1)$$

where $n_{X_i}$ is the number of amino acids of protein $X_i$ and $\rho_{\text{cell}}$ is the protein density in the cell.

Finally, we use the equations derived by Scott et al.[47] for their original proteome-partition model to predict the extent of gene-expression burden arising from the circuit, i.e. the reduction in growth

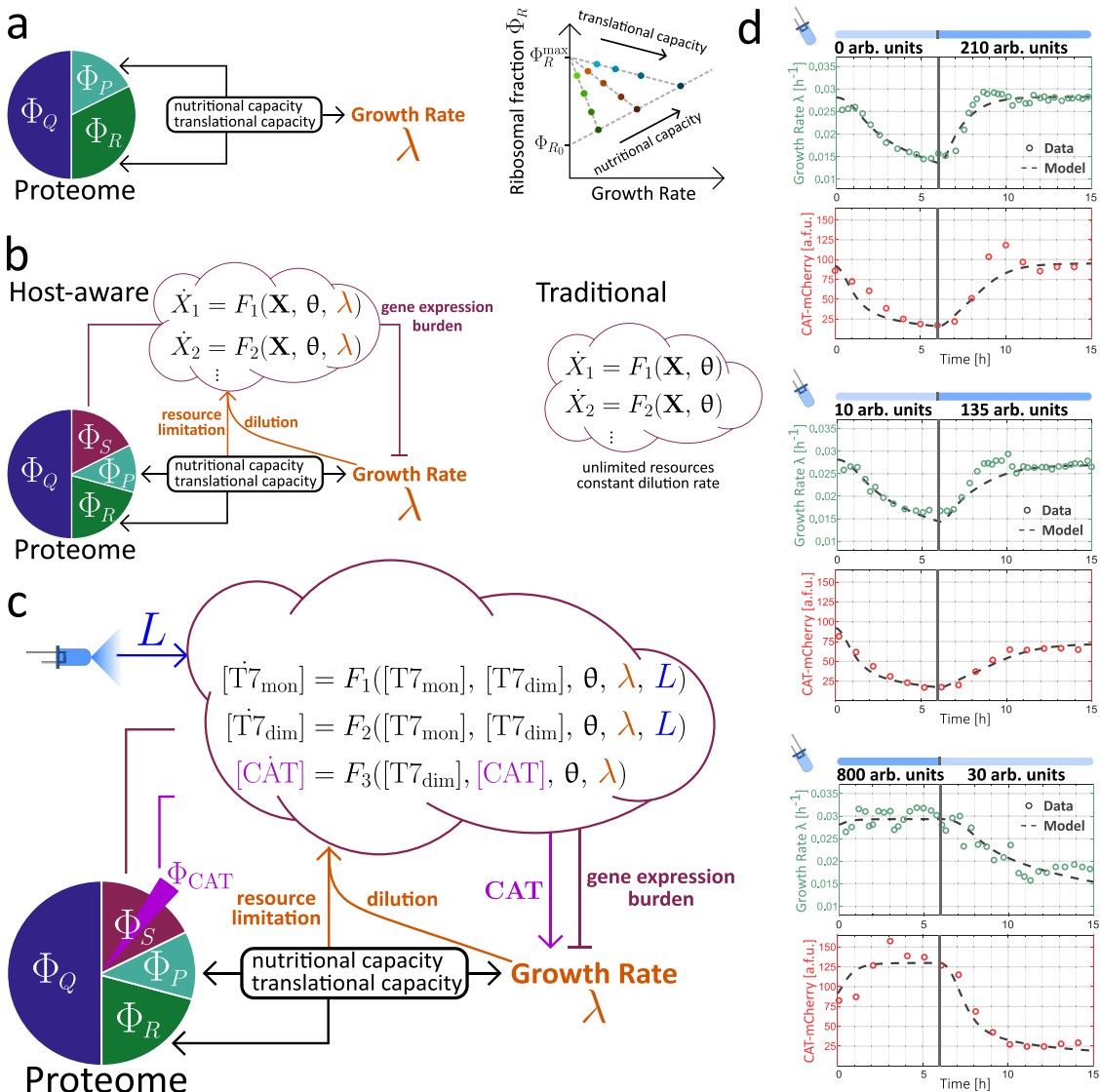

**Fig. 4 | Host-aware modeling framework applied to the photophilic strain. a** In proteome-partition models, empirical correlations between growth rate and ribosomal mass fraction arise from a balance between co-regulated sectors of the proteome: a fixed, house-keeping fraction ($\Phi_Q$) and two flexible sectors, the ribosomal fraction ($\Phi_R$) and the catabolic fraction ($\Phi_P$). The give-and-take regulation of the latter two sectors determines the cellular growth rate, mediating adaptation to environmental conditions characterized by nutrient quality and translational capacity. Figure adapted from refs. 47 and 54. **b** In our approach to host-aware modeling, the dynamics of arbitrary synthetic genetic circuits, described by a system of ODEs, are embedded into a proteome-partition framework that captures the physiological response of the cell. In contrast to a traditional ODE model, which does not consider the physiological adaptation of the host's growth rate, gene-expression burden caused by the circuit or limitations in the host's gene-expression resources, the host-aware framework seamlessly incorporates all of these host-circuit interactions without the need for extra free parameters. **c** Host-aware modeling framework applied to the photophilic strain. Blue-light intensity enters as an external parameter and the growth-modulating effect of expressing CAT in the presence of chloramphenicol introduces a direct interaction between the circuit and the host's growth rate. **d** Dynamic upshift and downshift experiments used to determine the best parameter values for the host-aware model of the photophilic strain. The parameterized model simultaneously recapitulates the dynamics of both cellular growth rate and resistance expression. Source data are provided as a Source Data file.

rate that results from the presence of the synthetic fraction of the proteome

$$\lambda(\Phi_S) = \left( \Phi_R^{max} - \Phi_{R_0} - \Phi_S \right) \frac{\gamma_0 \nu}{\gamma_0 + \nu} \qquad (2)$$

where $\Phi_R^{max}$, $\Phi_{R_0}$ and $\gamma_0$ are parameters that have been determined by Scott et al from detailed growth experiments (Fig. 4a) and $\nu$ is a parameter that characterizes nutrient quality and can be easily calculated from the growth rate of cells in the chosen experimental medium.

The final component of our modeling framework is a system of ODEs describing the circuit dynamics. It differs from a conventional model of a gene-expression network by explicitly considering the dependence of gene expression on the cellular machinery. We derive simple factors that capture how the availability of host transcriptional and translational resources scales with the growth rate and incorporate these to the production rates of the circuit components. The resulting equations for an arbitrary pair $m_{X_i}$ and $X_i$ of mRNA and protein species of the circuit ($i = 1, \cdots, u$) take the general form

$$\begin{aligned} \frac{dm_{X_i}}{dt} &= \omega_i \, T_i(\mathbf{m_X}, \mathbf{X}) \frac{\lambda}{\nu} + F_i(\mathbf{m_X}, \mathbf{X}) - \delta_i m_{X_i} \\ \frac{dX_i}{dt} &= \alpha_i m_{X_i} \lambda + G_i(\mathbf{m_X}, \mathbf{X}) - \lambda X_i, \end{aligned} \qquad (3)$$

where $\mathbf{m_X}$ and $\mathbf{X}$ are $u$-dimensional vectors containing the mRNA and protein species of the circuit and $\mathbf{T(m_X, X)}$, $\mathbf{F(m_X, X)}$ and $\mathbf{G(m_X, X)}$ are functions that describe all possible interactions between these species, at the transcriptional, post-transcriptional and post-translational levels. The factors $\frac{\lambda}{\nu}$ and $\lambda$, multiplying the transcription and translation rates, respectively introduce an explicit dependence on the host's transcriptional and translational machinery (See Supplementary Text Section 2 for a derivation of these terms). The parameters $\omega_i$, $\alpha_i$ and $\delta_i$ respectively refer to the transcription, translation, and mRNA-degradation rates of the $i$th species of the circuit. Growth-mediated dilution is absent from the mRNA differential equation because we assume that its effect is negligible in comparison to that of active degradation processes.

The collection of Eqs. (1)–(3), comprising our host-aware modeling framework, are based on the assumption that the growth rate equation, derived by Scott et al. for equilibrium conditions, holds in the—potentially non-equilibrium—context of a dynamic genetic circuit acting within the cell (see discussion for a comment on the validity of this assumption). Therefore, the model cannot be expected to provide quantitative predictions in all cases or to reflect an accurate mechanistic understanding. However, the approach is powerful, because it can be effortlessly incorporated into any existing ODE model of a synthetic circuit to provide intuition on the two-way interference between circuit and host physiology: The growth rate of the host, as a readout for physiological state, affects the circuit dynamics, both through dilution and by setting the availability of the gene-expression machinery, and the circuit impacts the host's growth rate through gene-expression burden (Fig. 4b). Furthermore, equations (1)–(3) contain the same number of free parameters as their conventional, non-host-aware counterparts that assume a constant dilution rate $\lambda$. This is due to the fact that the parameters that describe the host ($\Phi_R^{max}$, $\Phi_{R_0}$ and $\gamma_0$) were experimentally determined by Scott et al. for a range of common growth conditions[47]. As long as similar conditions are under consideration, all host parameters can be fixed and then the host-aware modeling framework captures interactions between circuit and host without increasing the risk of over-fitting.

### Host-aware modeling of optogenetic growth control recapitulates co-culture dynamics

We applied the host-aware modeling framework to the particular case of the growth-control circuit of the photophilic strain (Fig. 4c). For simplicity, we modeled the split-T7 as homodimers and focused on the dynamics at the protein level. The resulting system of ODEs, embedded in the context of *E. coli* through the use of our host-aware framework, describes how T7-monomers dimerize in a light-dependent reaction to produce active complexes that catalyze the production of the resistance. In the presence of a fixed external concentration of chloramphenicol, expression of CAT has a positive impact on the growth rate through the inactivation of intracellular antibiotic molecules. Model equations and derivations can be found in Supplementary Text Section 2.

To parameterize our model, we performed dynamic characterization experiments with the photophilic strain. We pre-incubated the cells under ambient light conditions before transferring the cultures to our turbidostat setup, where they were exposed to a precise illumination program. Our automated culturing and sampling capabilities allowed us to monitor both the growth rate and the resistance levels of the strain over time under both upshift and downshift conditions (Fig. 4d). We fitted simultaneously the growth rate and gene expression data with our model to determine the best choice of parameters. With this set of parameter values, the host-aware model is able to quantitatively reproduce the strain's dynamics both at the gene-expression and growth-rate levels with great accuracy. This means that we can simulate the dynamic trajectory of growth rate of the photophilic strain, $\lambda_p(L(t))$, in response to an arbitrary light input pattern $L(t)$.

Having developed an accurate model of the growth rate dynamics of the photophilic strain as a function of external illumination, we set out to model how these dynamics govern the composition of a co-culture in which both the photophilic and the constitutive strain are present. For this, we assume that the growth rate of the constitutive strain, $\lambda_c$, is time-independent and that the growth rate of the photophilic strain depends on time only through the applied dynamic light input. In such a case, the time evolution of the composition of the photophilic-constitutive co-culture can be characterized with a single ODE that describes the fraction of the photophilic strain, $\varphi_p$ (See Supplementary Text Section 2),

$$\frac{d\varphi_p}{dt} = \left(\lambda_p(L(t)) - \lambda_c\right)\left(1 - \varphi_p\right)\varphi_p. \tag{4}$$

Equation (4) has a simple geometric interpretation, which is depicted schematically in Fig. 5a. The system has two fixed points that correspond to dominance of either the photophilic ($\varphi_p = 1$) or the constitutive ($\varphi_p = 0$) strain. The stability of the fixed points is fully determined by the growth-rate difference between the strains, i.e. a winner-takes-all scenario, where the faster-growing strain eventually comes to dominate the co-culture, driving the other strain to extinction. The higher the disparity in growth rates, the faster the system converges to the stable fixed point.

We confirmed this experimentally by running co-culture experiments in open loop, i.e., with a fixed light-intensity input that does not feed back on the co-culture state (Fig. 5b and Supplementary Fig. 6). As expected, maximal light intensity lead to dominance of the photophilic strain, while the absence of light resulted in the opposite outcome. We observe faster convergence when the culture is incubated without light, which agrees well with the fact that the growth rates of the strains are more dissimilar in the dark than under maximal illumination (Fig. 2d). Finally, we simulated the composition of the co-culture in the open-loop setting by combining our parameterized model of the photophilic strain's growth rate with Eq. (4). The outcome matches quite closely the observed behavior of the co-culture (dashed lines in Fig. 5b), suggesting that the host-aware model of the photophilic strain that we developed, would allow us to investigate the behavior of the co-culture under optogenetic feedback.

### In silico optogenetic feedback stabilizes arbitrary strain ratios in co-culture

Equation (4) also reveals the way in which applying feedback control on the growth rate of the photophilic strain can stabilize the composition of the co-culture at arbitrary strain ratios. Co-existence between the strains is only possible if the growth rates are exactly the same. In that case, Eq. (4) becomes zero independently of the composition of the co-culture, meaning that the current ratio of strains becomes locked as a fixed point. It is worth noting, however, that even the smallest deviations from perfectly matched growth rates would cause the system to revert back to a winner-takes-all scenario, where the strain ratio slowly shifts towards the stable fixed point. Therefore, the role of feedback is twofold: First, the input light levels steer the co-culture composition towards the desired state by modulating the difference between the growth rates of the strains. Then, after the desired setpoint has been reached, the light input must ensure that the growth rates remain matched, reacting to the unavoidable fluctuations in growth rate that would cause the strain ratio to move away from its setpoint value.

To achieve these tasks through optogenetic feedback, we decided to determine the necessary light inputs via a PID controller (Fig. 6a), a long-standing control strategy that has found many applications in industry because of its simplicity, performance, and versatility. The integral component of a PID controller ensures that the closed-loop system can track constant setpoints, while the proportional and derivative components optimize the transient dynamics[55]. In order for the

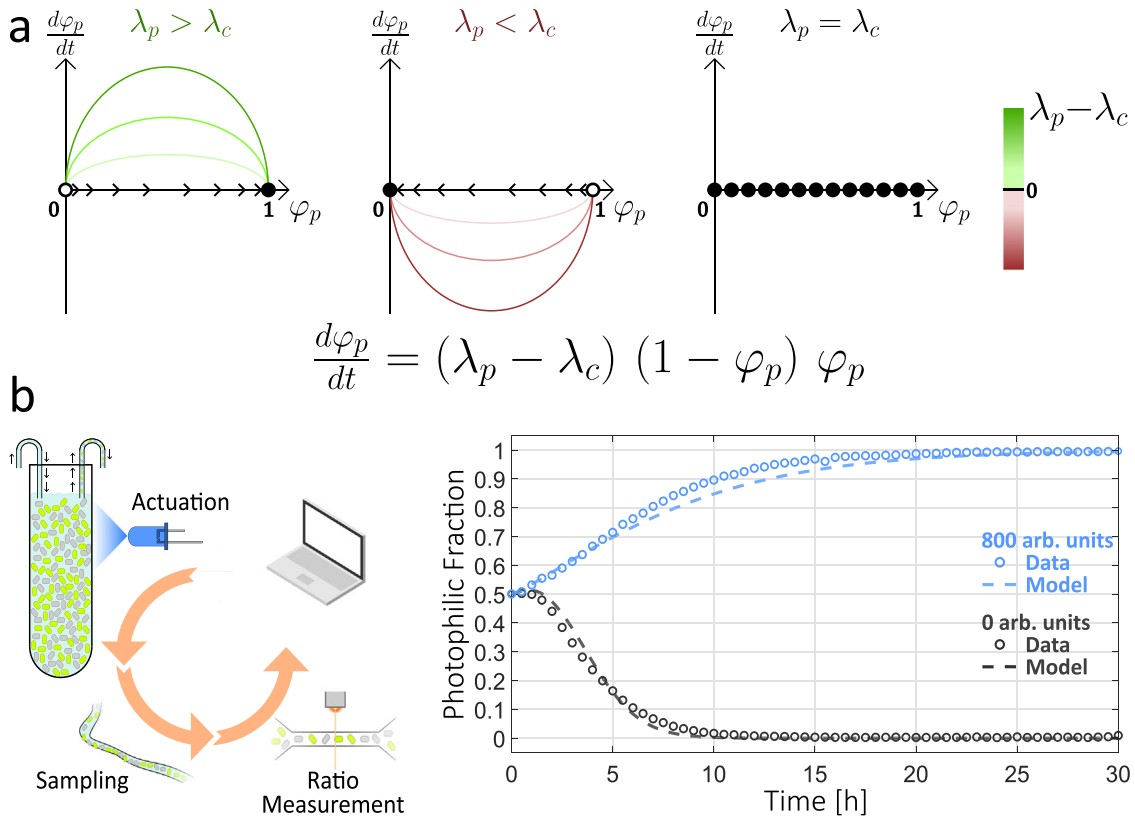

**Fig. 5 | Open-loop dynamics of the photophilic-consitutive co-culture. a** The composition of the photophilic-constitutive co-culture can be characterized by the photophilic strain fraction $\varphi_p$, whose dynamics obey the depicted ODE. Phase portraits illustrate three qualitatively distinct scenarios. Full and empty circles denote stable and unstable fixed points, respectively, and trajectories evolve in time following the flow denoted by the arrows. In all cases, the sign and magnitude of the growth rate difference, $\lambda_p - \lambda_c$, determine the fate of $\varphi_p$ and the speed of convergence to equilibrium. If $\lambda_p > \lambda_c$, the photophilic strain inevitably dominates: $\varphi_p = 1$ (Left). If $\lambda_p < \lambda_c$, the constitutive strain dominates: $\varphi_p = 0$ (Center). If the strains grow at equal pace (Right), any value of $\varphi_p$ can be a fixed point of the dynamics. **b** (Left) Schematic of open-loop case. A constant light intensity is delivered throughout the experiment and samples are collected periodically to monitor the co-culture composition. (Right) Experimental behavior of the co-culture in an open-loop setting, illustrating the three qualitative cases from (**a**). Dotted lines represent computational simulations of the equation in (**a**), where $\lambda_p = \lambda_p(L(t))$ is determined dynamically from the delivered input profile $L(t)$ with a host-aware model of the photophilic strain. Source data are provided as a Source Data file.

closed-loop system to be stable and to have appropriate dynamics, one must find a set of gains for the three controller components that result in optimal controller performance. Using our model of how the co-culture responds to arbitrary light inputs, we carried out a computational screen to find the best-possible gains for our controller (Fig. 6b). We randomly sampled sets of gains and simulated the behavior of the closed-loop system for a couple of desired setpoints of strain ratios. To each simulation, we assigned a score that penalizes deviations from the optimal trajectory. Through this simple procedure, we were able to find a set of gains that was predicted to result in fast convergence to the setpoint, no overshoot, and only slight oscillations at equilibrium. We chose this over alternative sets of optimal gains, such as one which would produce no oscillations at equilibrium but a higher overshoot and a longer transient (Supplementary Fig. 7). This highlights the fact that it is possible to choose different optimization targets depending on the application. The importance of this computational screening is illustrated by the fact that randomly chosen controller gains resulted in prohibitively long transients or widely unstable systems (Supplementary Fig. 8).

We used the optimal set of gains to implement the optogenetic-feedback loop schematized in Fig. 6a. The two strains are inoculated at a defined initial ratio and then samples are collected automatically every thirty minutes and passed through a flow cytometer to monitor the composition of the co-culture in real time. In each sampling step, after evaluating the difference between the current strain ratio and the desired setpoint, the PID controller updates the intensity of the blue-

light LED so as to steer the trajectory of the co-culture composition to the defined goal. Once initialized, the entire closed-loop experiment runs autonomously without the need for human intervention. Moreover, the multiplexed capabilities of the *evotron* framework (Fig. 3) allow us to control up to five independent co-culture vials, each with its own behavior objective.

The experimental trajectories of our co-cultures exhibited an extraordinary resemblance to the simulations, demonstrating again the quantitative predictive power of our host-aware model. Starting from an initially balanced composition, we could drive the strain ratio to arbitrary values, establishing dominance of either the photophilic or the constitutive strain (Fig. 6c–d, Supplementary Figs. 9 and 10). Furthermore, the desired ratios were stabilized by the feedback for up to 40 h, or the equivalent of around 80 bacterial generations in our culturing conditions. After this, the co-culture composition started to slowly drift towards a state of dominance of the constitutive strain, in spite of the counterbalancing efforts of the controller, probably reflecting the fixation of escape mutations in the population of one or both co-cultured strains (Supplementary Fig. 11). Nevertheless, before the onset of this escape phenomenon, the desired composition was stably maintained with only minor oscillations around the setpoint, a slight instability that our model suggests could be mitigated by increasing the sampling frequency (Supplementary Fig. 12).

Finally, we show that we can also successfully steer the co-culture to track a changing setpoint, such as first converging to a state of dominance of the constitutive strain, then switching to the opposite

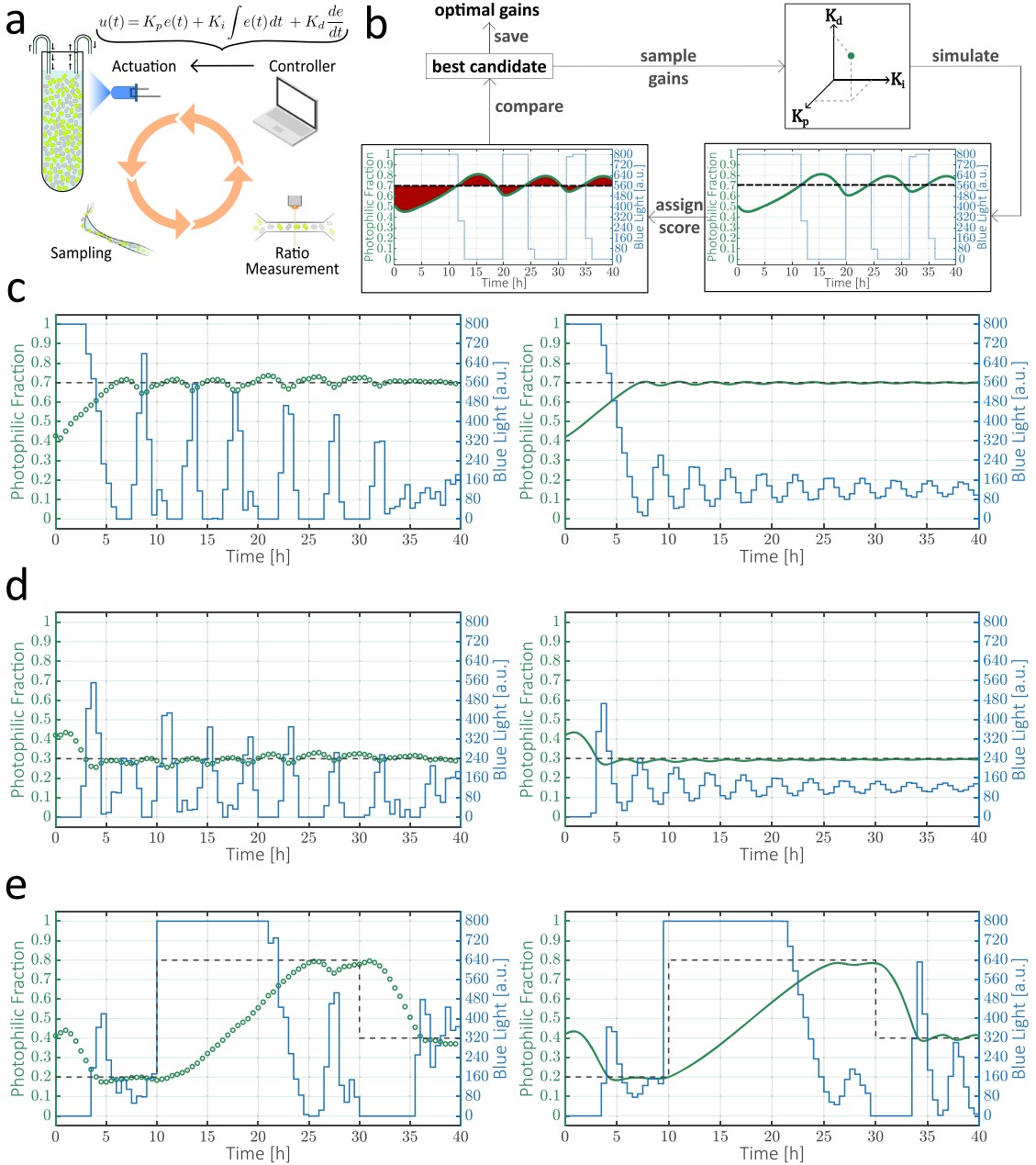

**Fig. 6 | Computational screening for optimal controller parameters and closed-loop control of co-culture composition. a** Schematic of closed-loop control experiment with a PID controller. **b** Computational screening for optimal PID gains. Sets of randomly sampled gains ($K_p$, $K_i$, $K_d$) are used to simulate the expected trajectory of the co-culture in a closed-loop setting with a defined target setpoint. Trajectories are scored according to their total deviation from the target strain ratio in the relevant time window. **c–e** Closed-loop control of co-cultures with the same initial strain ratio and different target setpoints. Both model predictions (Right) and experiments (Left) are shown. **c** Target photophilic fraction: $\varphi_p^{\text{set}} = 0.7$. **d** Target photophilic fraction: $\varphi_p^{\text{set}} = 0.3$. **e** The strain ratio is forced to track setpoints that change from a target photophilic fraction of $\varphi_p^{\text{set}} = 0.2$ to $\varphi_p^{\text{set}} = 0.8$ ($t = 10$ h) and to $\varphi_p^{\text{set}} = 0.4$ ($t = 30$ h). Source data.

composition and afterwards moving to a balanced ratio of strains (Fig. 6e and Supplementary Fig. 13). As a whole, these results demonstrate that the composition of a naturally unstable bacterial co-culture can be arbitrarily and accurately controlled for a reasonably long time window using our optogenetic feedback approach. This opens up the possibility of implementing dynamic programs that could find applications in both consortium-based bioproduction and the study of ecological interactions within microbial communities.

## Discussion

In this work, we demonstrate dynamic control over the composition of a two-strain bacterial co-culture. In a fully automated platform, we implement feedback control on the growth of a single strain through a combination of optogenetics and external light inputs. The accuracy and fast dynamics of our control method showcase the advantages of optogenetics over conventional, chemical induction, allowing instantaneous delivery and removal of inputs and fine control over dosage. In agreement with a recent study that stabilizes a pair of strains by balancing amensalism with competitive exclusion[35], engineering a single strain was enough in our case to provide good dynamic control over the composition of the two-strain community. Here, we chose to work with the blue-light inducible opto-T7 polymerase, because of its fast dynamics and its compact design that doesn't require the expression of co-factors[51]. Controlling the composition of more diverse

communities, however, would require the use of orthogonal optogenetic tools to modulate the growth of more than one strain. Although orthogonality has been demonstrated for a pair of green-red systems[56], both of these systems absorb in the blue spectrum and are thus incompatible with blue-light tools. In practice, the availability of reliable optogenetic tools that respond to orthogonal wavelengths could limit the scalability of our approach to more diverse communities. Nevertheless, introducing multiplexing methods[57] and focusing external control on key driver species within interacting communities[58] could provide a way forward.

Previous attempts to stabilize microbial co-cultures have relied largely on self-limiting populations[29,30], harnessing phenotypic switches in the co-cultured populations[59–61], engineered interactions between strains and emergent behaviors[6,21,31,32,34]. Methods for ensuring co-existance in a chemostat through control of the dilution rate have also been proposed[62]. Although these approaches can generate a richness of equilibrium and dynamic behaviors, they often sustain only a limited set of strain ratios and community dynamics that are predetermined by intrinsic properties of the system. Our combination of external feedback and single-strain control, in contrast, enables us to stabilize the community at any desired strain ratio and to easily change the setpoint during an experiment. Furthermore, the fact that the controller runs on a computer means that we can implement arbitrary control strategies tailored to the specific dynamical objectives, without the constraints of chemical or interspecies interactions.

Our approach for controlling growth, the optogenetic expression of an antibiotic-resistance gene via an opto-T7 polymerase[51], resulted in both longer genetic stability[41] and faster dynamics[38] than previous implementations of optogenetic growth control. However, our method still suffered from the eventual fixation of escape mutations after around 80 generations, which limits the time window in which we can maintain compositional control over the community. Loss-of-function mutations are inevitable in circuits that reduce the fitness of the host, although some strategies have been recently proposed for improving genetic stability and delaying the onset of mutations[63,64]. However, it is worth noting that the use of a real-time feedback can, in principle, prolong the stability of the system by adapting the light-input profile to counteract the effect of certain mutations, a behavior we likely observe in our closed-loop experiments after around 35 h (Fig. 6c–d, Supplementary Figs. 9 and 10). If the effect of mutations is known, adaptive control strategies such as gain scheduling could be implemented to counterbalance the altered properties of the community. Further potential drawbacks of our approach to growth control are the obligate use of antibiotics and continuous culturing methods, both of which might not be desirable in large-scale industrial settings. However, we were able to obtain good results with concentrations of chloramphenicol ten times lower than those used for selection. Furthermore, the overall strategy of using in silico feedback coupled to optogenetic growth control does not depend on the specific details of how growth is affected by light. Therefore, the same strategy could be applied to the optogenetic expression of other growth-modulating factors, such as bottleneck metabolic enzymes[41], RNA-polymerase[65] or growth-inhibiting toxins[30], should these be more favorable to the application at hand.

We also use host-aware modeling to accurately predict the dynamics of the bacterial community and to properly tune the parameters of our controller. A priori computational optimization was responsible for the excellent performance of our closed-loop control. Furthermore, it considerably reduced the final experimental effort, which would have been considerably larger with a trial-and-error approach. Both the simplicity and success of our procedure suggest that it could be replicated for arbitrary applications, such as adding biosynthetic modules on top of the growth control circuits for cooperative bioproduction[20,24,25,66]. Although such modifications would affect the growth rates of the strains, the same model-guided

optimization procedure could be used to adapt the controller gains to the altered community dynamics. The use of a host-aware model is not stringently required for a satisfactory performance of the closed-loop system. For example, simpler, phenomenological models that fit the open-loop response of the community could also be used for the computational optimization of PID gains, simplifying the implementation of our compositional-control platform even further. Given that model-predictive control schemes can cope with model inaccuracies to a large extent, both types of models could also be used to implement such schemes, which would potentially outperform PID control for time-varying reference signals[41]. However, we note that harnessing the controlled co-culture for biotechnological applications would require the introduction of further genetic loads (e.g., biosynthetic pathways or genetic circuits) that would have a–potentially convoluted–impact on the growth rates of the strains. Host-aware models might be more flexible when it comes to accommodating such changes. Moreover, the host-aware framework presented here could be relevant for applications beyond the scope of microbial community control.

Our host-aware modeling framework consists of minor, phenomenologically motivated modifications to conventional ODE models of gene expression. Therefore, it can be easily applied to existing models of arbitrary genetic circuits. One problematic assumption of our framework is that the laws governing the allocation of the proteome can be extrapolated to non-equilibrium conditions, such as the transient phase of a circuit's dynamics. However, the lack of any appreciable delay between the change in resistance levels and its effect on the growth rate (Fig. 3c) suggests that growth-related adaptation might indeed be considered to be in quasi-steady state in relation to the protein dynamics of the circuit, at least in our particular case, where growth is modulated by varying the levels of CAT in the presence of chloramphenicol.

Many modeling strategies have been developed in recent years to explain unintuitive circuit behaviors that arise from resource-sharing between exogenous gene networks and host physiology[67–71], some of which are based on the same type of course-grained description of the bacterial proteome[67,70] that we use in our framework. In contrast to these approaches, we do not base our model on mechanistic representations of the processes underlying host-circuit interactions, which might result in lower quantitative power and limited interpretability. Nevertheless, we observed good quantitative agreement between our model predictions and data from dynamic experiments. This is particularly remarkable given that our circuit explicitly modulates cellular growth and is, therefore, expected to have a large impact on the physiology of the host and the abundance of gene-expression resources. We hope that the simplicity of our modeling approach and the fact that host-circuit interactions are introduced without the need for extra parameters, will make our framework appealing to the community. Even though quantitative predictions might not always be attainable, our framework could provide a quick and easy way of testing whether the qualitative behavior of a circuit is expected to change when introduced into the complex context of an E. coli host.

In achieving our aim of dynamic compositional control in bacterial co-cultures, we also highlight the potential of lab automation efforts by devising evotron, a generic framework for fully automated, high-throughput continuous cell culture, sampling, and light stimulation. We enhanced the scope of the versatile eVOLVER platform[53] by introducing an optogenetic stimulation functionality and by improving the on-board cell density measurement. At the same time, we also expanded the adaptable accessories for the widely used Opentrons OT-2 lab automation robot to facilitate automated high-frequency sampling from cell culture platforms directly placed on its deck. We believe that our modular, automated framework can be easily adapted and employed for other types of studies, reducing the barrier in achieving lab automation by relying on minimal and readily available resources.

Recent years have witnessed an ever-increasing interest in studying the dynamics and interactions within microbial communities in the natural world, as well as in reaping the advantages of microbial consortia for industrial purposes. However, many studies and proposed applications are still hindered by the inability to stabilize co-cultures of microorganisms growing at different rates. Our study serves as a proof of principle that external feedback strategies can be a valuable tool in this context, providing accurate control over both the dynamics and the precise composition of a simple co-culture. We hope that our approach will enable applications that unlock the full potential of synthetic biology for implementing complex programs in heterogeneous communities.

## Methods

### Growth conditions

All experiments were performed in LB supplemented with 0.1% Tween 20. Tween 20 was added to experiment media to reduce the amount of biofilm formation on the glass vials of the eVOLVER and also to overnight and pre-cultures for consistency. Antibiotics (Sigma-Aldrich Chemie GmbH) were used at the following concentrations: spectinomycin, 50 µg/mL; ampicillin, 100 µg/mL; chloramphenicol, 3.4 µg/mL. Overnight cultures contained only ampicillin and spectinomycin. Media used for pre-cultures and *evotron* experiments also contained chloramphenicol.

For overnight and experiment pre-cultures, cells were grown in 5 ml of media in 14 ml transparent polypropylene tubes (Greiner) at 37 °C in an environmental shaker (Excella E24, New Brunswick) set to 230 rpm. In the eVOLVER, cells were grown in 20 ml LB with medium stirring (eVOLVER level 8) and temperature control set to 37 °C. Turbidostat regulation was set to keep the cultures at optical densities ($OD_{600}$) between 0.10 and 0.15, regularly switching the OD setpoint between the upper and lower limits in order to obtain periodic segments of increasing OD from which to determine the growth rate of the culture.

After the Opentrons OT-2 robot was set up with eVOLVER replacing its deck, it was covered with black foil and tape to shield the cultures from ambient light. We observed sudden growth defects after ~10–15 h, which we attribute to a drop in oxygen levels within the Opentrons chamber in which the turbidostat cultures are located (Supplementary Fig. 14). This was confirmed by the fact that normal growth was restored after adding an external supply of pressurized air to the interior of the Opentrons (Supplementary Fig. 15). The air supply was kept running at 4bar during turbidostat experiments.

In addition, in spite of the supplemented detergent, we observed biofilm accumulation on the glass vials of the eVOLVER over time. This process occurred irrespective of the illumination conditions and after 8–12 h began interfering with the OD sensor readings, leading to a spurious increase in the estimated growth rate. To prevent this from happening, we manually transfer cultures to fresh, pre-warmed vials every 6–7 hours, including clean stir bars. Exchanging vials with this frequency prevented biofilm accumulation while having no noticeable effect on cell growth.

### Strains and plasmids

We used *E. coli* strain BW25113 as host for the plasmids of the constitutive strains. For the photophilic strain, a constitutive mVenus cassette was additionally inserted into the chromosome of BW25113 to make the strain yellow fluorescent (BW25113 attB::venus). λ-integration plasmid pSKA637 was constructed via isothermal assembly[72] with an mVenus sequence amplified from pZS2-123[73] using oligos oSKA826 (GAGAAATCAAATTAAGGAGGTAAGATAATGAGCAAAGGTGAAGAAC)/oSKA827 (GTTTTTTTGCGCTCTAGTATCATTATTTATACAGTTCGTCCATACCG) and the lambda burden monitor[23] backbone amplified using oligos oSKA822 (CATTATCTTACCTCCTTAATTTGATTTCTC)/oSKA823 (TAATGATACTAGAGCGCAAAAAAC) and cloned into C

C118(λpir)[74]. The resulting plasmid pSKA637 is a Venus YFP version of the original sfGFP lambda burden monitor. pSKA637 was integrated into the chromosome of BW25113 at the attB site using λ-integrase plasmid pInt-ts as described in ref. 75. The resulting strain, SKA1515 (BW25113 attB::venus), was sequence-verified and is kanamycin-resistant.

All plasmids were constructed from a custom-made library of parts with optimized overhangs[76] using standard Golden-Gate assembly methods and modular cloning (MoClo)[77] with restriction enzymes BsaI and BbsI (New England Biolabs). To make the photophilic strain (bJAG132), plasmids pAB276 (pSC101, AmpR, Opto-T7) and mJAG063 (p15A, SpecR, CAT gene under control of a T7 promoter), were transformed into SKA1515. pAB276 was derived from pAB150[51] by exchanging the original chloramphenicol resistance marker with an ampicillin resistance marker. To build mJAG063, we modularly combined parts containing the consensus sequence of the T7 promoter, the original 5'UTR from pAB050[51], the CAT coding sequence fused C-terminally (short linker: GGGSGGGS) to an mCherry sequence truncated at its N-terminus by 10 amino acids and the strong synthetic terminator L3S2P21[78] to build a transcriptional unit in a p15A backbone with a spectinomycin resistance marker.

To build the constitutive strains, plasmid mJAG090 (colE1, AmpR, constitutive mCherry cassette) was transformed into BW25113 to obtain the ampicilin-resistant strain bJAG234. This strain was the basis for all constitutive strains. To obtain different levels of constitutive growth, several plasmids (pSC101) were cloned, which express CAT at different levels due to different ribosome-binding sequences and the presence or absence of a degradation tag. These plasmids were transformed into bJAG234 to obtain the constitutive strains used in this study. See Supplementary Table 2 for an overview of the strains and plasmids used in this study.

The sequences for the plasmids used in this study are available in Supplementary Text Section 3.

### Experimental details

Characterization and co-culture experiments were performed in the following way. Overnight cultures were inoculated from glycerol freeze stocks in 5ml LB supplemented with ampicillin, spectinomycin, and 0.1% Tween 20 (without chloramphenicol). The following day, the cultures were diluted into fresh experiment media (with chloramphenicol) to start a pre-culture at around 0.03 OD and incubated for around 2:30 h at 37 °C with shaking in an environmental shaker with transparent lid, i.e. exposed to ambient light. In the case of co-culture experiments, the cultures were mixed at this point to obtain the desired initial strain ratios. After this, the cultures were diluted in 20 ml of experiment media in the eVOLVER glass vials to a starting OD of 0.075 and transferred to the *evotron* platform to start the automated experimental pipelines. The maximal light intensity used in this study was 800 arb. units, which did not result in any appreciable toxicity on cell growth (Supplementary Fig. 16). Experiments were carried out in biological triplicates, either on different days or from separate independent cultures on the same day unless explicitly stated. Light levels throughout this paper are given in arbitrary units of digital intensity levels (arb. units). For a conversion of these digital-level arbitrary units to the power of the LED light measured at the center of an eVOLVER sleeve, see Supplementary Fig. 17.

### Flow cytometry

In our experiments, fluorescence measurements were performed on a Cytoflex S (Beckman Coulter) flow-cytometer running on CytExpert v2.4 software integrated with our *evotron* framework. mVenus was measured with a 488-nm laser and 525/40 bandpass filter, and mCherry was measured with 561-nm laser and 610/20 bandpass filter. The gain settings were as follows: forward scatter 100, side scatter 100, GFP 500, PE 145, mCherry 500. Gating of flow-cytometry events was

performed as follows. Both in automated closed-loop experiments and in the custom python scripts used for analysis of the data from characterization experiments, first a polygon gate was applied on the FCS-H vs. SSC-H channels to select for living cells (P2). P2 was then further gated in the SSC-Width vs. FSC-H channels to select for single cells (P4). In co-culture experiments, the P4 population was further separated into two subpopulations (corresponding to the photophilic and constitutive strains) by applying a fixed threshold gate on the GFP-A channel (Threshold = 6500) (Supplementary Fig. 18).

### *evotron* platform construction and preparation
Details related to the assembly and construction of the *evotron* framework are mentioned in Supplementary Text Section 1. Different components used in the framework are as follows:

1. eVOLVER platform[53] version 2.0 (Fynch Bio) with modified sleeves (Fig. 3a).
2. OT-2 Robot (Opentrons); Server version: 4.3.1; Firmware Version: v1.1.0-25e5cea; Protocol API version: 2.0.
3. Arduino Mega 2560 and USB to TTL serial cable (TTL-232R-5V).
4. Peristaltic pump (HyperCyt, Intellicyt Corporation).
5. Standard stainless steel non-coated tip (Ref 30032172, Tecan) as sampling-needle.
6. Silicone tubing 1.5 mm ID × 3.00 mm OD.

A custom-developed OT-2 calibration routine was performed every 3 months to determine the location of different eVOLVER sleeves and cleaning solution bottles on the OT-2 deck. These location data were then entered into the main protocol code running on the OT-2.

### *evotron* software development
The primary software routine for the control computer was developed and executed on a MATLAB R2021a framework. Cytoflex measurement acquisition code was custom developed in the .NET framework 4.5.2 environment using Microsoft Visual Studio 2017 IDE. The main protocol code running on the OT-2 was written and executed in Python v3.7.1. The controller code running on Arduino mega 2560, and LED stimulation control code running on Arduino SAMD21 (UC2 Arduino: Supplementary Fig. 19b) were developed, compiled and uploaded using Arduino IDE v1.8.12. OD regulation, growth rate calculation, and custom-developed LED stimulation-intensity control code for the eVOLVER platform were run on Python v3.6.8.

### Host-aware modeling, simulations, and optimization of PID controller gains
Details on our host-aware modeling framework and its application to model the photophilic strain, as well as the parameter values used for simulations are provided in Supplementary Text Section 2 and Supplementary Table 1 respectively.

Simulations of the photophilic strain and co-culture dynamics, as well as the optimization of the PID gains were carried out using custom MATLAB R2019b (Mathworks) scripts. Our implementation of PID control includes an anti-windup scheme that relies on the back-calculation method[55], which can be tuned via a fourth gain ($K_{bc}$) that we optimized alongside the gains of the proportional, integral, and derivative components. The final gains used in our closed-loop experiments are: $K_p = 5.9055 \times 10^3$, $K_i = 3.0382$, $K_d = 2.3427 \times 10^5$, $K_{bc} = 0.01 \times K_i$.

ODEs were integrated using MATLAB's stiff solvers ode15s and ode23s. The optimal value of the nutritional capacity $\nu$, which describes the quality of our LB media with supplemented detergent and antibiotics, was determined manually so that the predicted growth rate of the strain in the absence of chloramphenicol matches experimental observations (Supplementary Fig. 16). The remaining parameters for the model of the photophilic strain were either taken from the literature or determined by fitting the model simultaneously to

dynamic growth and gene expression data from three up- or downshift experiments using the MATLAB's non-linear least-squares solver lsqnonlin. In these experiments, the cultures have been exposed to ambient light previous to the start of the automated experiment. Therefore they all start at a similar growth rate, before adapting to the light condition in the first part of the up- or downshift experiment (Fig. 4d). We model this by introducing a separate parameter, $L_0$, which describes the blue-light level that would correspond to ambient light exposure. We begin by modeling the pre-culture phase of the experiment, i.e. we consider the photophilic strain subject to a constant light intensity and determine the value of $L_0$ as the light intensity that recovers the growth rate observed at the start of the *evotron* experiment. Afterwards, we keep $L_0$ fixed and determine the rest of the parameters by fitting the up- or downshift experiments through simulations that contain three phases: a pre-culture phase until equilibrium with a constant light input ($L_0$); a shift to the first light intensity of the experiment, which lasts six hours; a second shift to the second light intensity for the remainder of the experiment. Therefore, our up- and downshift experiments effectively contain two phases of transient dynamics, the adaptation from ambient light ($L_0$) to the light levels of the first phase of the experiment ($t < 6$ h) and the adaptation phase after the light shift ($t > 6$ h).

The MATLAB scripts used to run the simulations presented in this paper are available in the Supplementary Software file.

### Data analysis and visualization
Data obtained from *evotron* experiments were analyzed and plotted using custom python or MATLAB (Mathworks) scripts. Plots were then formatted and brought together to form the paper figures using Inkscape (v0.92, open source). Cartoons and schematics were also made using Inkscape.

### Reporting summary
Further information on research design is available in the Nature Research Reporting Summary linked to this article.

## Data availability
Data needed to evaluate the conclusions in the paper are present in the paper and/or the Supplementary Materials. Source data are provided with this paper.

## Code availability
All *evotron* related software codes and routines are available in GitHub repository: https://github.com/santkumar/evotron.git (Zenodo DOI 10.5281/zenodo.6908131). The MATLAB scripts used to run the simulations presented in this paper are available in the Supplementary Software file.

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

## Acknowledgements

We thank Dr. Stephanie Aoki and Dr. Armin Baumschlager for help with strain design and useful discussions, Dr. Brandon Wong and Dr. Gregor Schmidt for help with the development of the *evotron* platform and Paul Argast for providing hardware workshop services. We also thank Dr. Stephanie Aoki and Dr. Maurice Filo for proof reading the manuscript. This project has received funding from the European Research Council (ERC) under the European Union's Horizon 2020 research and innovation programme (CyberGenetics; grant agreement 743269) and from the European Union's Horizon 2020 research and innovation programme (COSY-BIO; grant agreement 766840).

## Author contributions

J.G. conceptualized the study, built the synthetic *E. coli* strains, performed the characterization experiments, devised the host-aware modeling framework and performed the computational modeling and optimization of PID gains. S.K. developed the *evotron* framework, modified the culturing setup for optogenetic stimulation and wrote the central code for running the automated platform with closed-loop control. J.G. and S.K. performed the experiments for optimization of the automated setup and the co-culture experiments. J.G. and S.K. wrote the manuscript. M.K. secured funding and supervised the project.

## Competing interests

The authors declare no competing interests.
