## [Peer Review File · Nature Communications]

Reviewers' Comments:

Reviewer #1:

Remarks to the Author:

In their manuscript Gutiérrez, Kumar, and Khammash demonstrate a novel and innovative optogenetic approach to cybergenetic control of bacterial co-cultures. Their work delivers insight and solutions for several significant challenges in the field: how to implement controllable community dynamics with Synthetic Biology; how to model these communities and translate models to practice; and how to establish high-capability automated experimental pipelines for their analysis and development. Each of these topics is approached with robust analysis and integration of past work in the field, while delivering novel techniques which may have far-reaching applications, highlighting the rigor and creativity of the authors' approach. Furthermore, methods and results are all explained exceptionally clearly and with strong attention to detail, making the paper a valuable resource for the others in the field hoping to build upon the authors' innovations in the future. Overall I found the paper a very engaging and informative read: I therefore strongly recommend it for publication and congratulate the authors on their excellent work!

Below are some comments on specific parts of the text and analysis, mostly quite minor, which I hope the authors may consider in their revision.

Line 39 – The definition of the “Competitive exclusion principle” might be updated here, in particular to specify that an important tenet of the principle is that competitors have the same/similar resource requirements. That is, a community without direct stabilizing interactions may support multiple concurrent species so long as their nutrition/resource requirements are sufficiently dissimilar. I note that the caption of Figure 1A subsequently defines the competitive exclusion principle with this requirement (i.e. competition for common resources). If this change is made perhaps line 41 could be toned down somewhat, e.g. that it can be “difficult” to maintain community composition (rather than impossible).

Line 50 – Somewhere in this paragraph references to DOIs 10.1002/biot.202100169 and 10.1038/s41565-021-00878-4 could be added. These studies demonstrate some attempts at computational control of a cellular communities (admittedly not realised as elegantly with optogenetics, and with a much more limited scope and control/modelling technique compared to the present study).

Line 171 – The detailed description of the evotron setup is much appreciated. Do you have any sense of how it compares to the recently published ReacSight toolbox (DOI 10.1038/s41467-022-31033-9)? A detailed discussion of this comparison is likely not essential to the paper's core message, however perhaps readers would appreciate a reference to this background/alternative should they aim to implement a similar approach in their own work.

Line 307 – typo “... same number of free parameters AS their conventional...”

Line 313 – Is there a reason this figure is referenced as “A (Left)...” etc rather than A,B,C,D...? I will leave it to the authors' final judgement, but perhaps it is easier and faster for the reader to interpret if each sub-panel is conventionally labelled with a letter.

Line 365 – Axis labels on the small graphs in Panel B were small/difficult to read when printed. Font size could be increased here?

Line 405 – Apologies if this stated somewhere earlier that I missed; why is the evotron framework only able to control five independent vials in this case? My understanding was the eVolver platform had capability for sixteen vials in parallel. Would it be possible to comment on the limiting factors in the evotron approach's throughput (is it due to time required for each sample and subsequent cleaning?). Perhaps this could be discussed in Supplementary Section 2 where evotron is described in more detail.

Line 408 / Line 465 – Here (and elsewhere) the importance of the a priori host-aware modelling is stressed. E.g. on line 465 that it was “responsible for the excellent performance of our closed-loop

control. Furthermore, it considerably reduced the final experimental effort,". I agree completely with the authors' assessment that this host-aware framework seems effective and easy-to-implement. However, it is difficult to justify that this was the one factor responsible for the excellent performance of the closed-loop control. For example, if you had taken some open-loop step responses (e.g. as in Fig 4D) and fit a simple one-ODE system to this, could this have been used to computationally tune a PID with (roughly) similar outcome (and even less effort)? This question seems important for the target audience of the paper (e.g. those working in control application fields as outlined in the Discussion of the manuscript), who may wish for an as-simple-as-possible methodology that achieves reasonable outcome. Perhaps the authors could comment on the potential pitfalls of using a simpler modelling framework in PID tuning – in what situations might it be (or not be) good enough? Perhaps this could include some commentary on how such modelling approaches might perform should a MPC approach be used rather than PID (as done in some of the authors' past work on cybergenetics control).

Congratulations again to the authors on their excellent work!

Harrison Steel

Reviewer #2:

Remarks to the Author:

This manuscript details work that allows one to externally control the ratio of two bacterial strains growing in the same turbidostat. To do this, the authors use a PID controller to measure and adjust the growth rate of one of the two cell types. The growth rate is adjusted using a light-sensitive split T7 RNAP that regulates the expression of chloramphenicol acetyltransferase, an enzyme that confers resistance to chloramphenicol. Overall, I think this is pretty stellar work and should be of interest to many in the fields of synthetic biology and metabolic engineering.

I usually would now list all the faults I find with the paper, but, in truth, I didn't really find anything glaring. The experiments are done well and the appropriate controls look solid. The control theory is also very well explained and should be reproducible. I might give it a try in my lab if I can get my people to tinker with our turbidostats.

Also, when I first read the paper, I had thought Mario Di Bernardo had done almost exactly this using microfluidics. But looking back, I can't find it, just the theory paper his group did in ACS Syn Bio.

Nicely done!

Matthew R. Bennett
Rice University

Response to the reviewers' comments

We would like to thank both reviewers for their kind and supportive comments, as well as for all their suggestions, which allowed us to improve the quality of the manuscript.

Reviewer #1 (Remarks to the Author):

In their manuscript Gutiérrez, Kumar, and Khammash demonstrate a novel and innovative optogenetic approach to cybergenetic control of bacterial co-cultures. Their work delivers insight and solutions for several significant challenges in the field: how to implement controllable community dynamics with Synthetic Biology; how to model these communities and translate models to practice; and how to establish high-capability automated experimental pipelines for their analysis and development. Each of these topics is approached with robust analysis and integration of past work in the field, while delivering novel techniques which may have far-reaching applications, highlighting the rigor and creativity of the authors' approach. Furthermore, methods and results are all explained exceptionally clearly and with strong attention to detail, making the paper a valuable resource for the others in the field hoping to build upon the authors' innovations in the future. Overall I found the paper a very engaging and informative read: I therefore strongly recommend it for publication and congratulate the authors on their excellent work!

We thank this reviewer deeply for his positive appreciation and support, as well as for his comments, which helped us improve the quality of the paper!

Below are some comments on specific parts of the text and analysis, mostly quite minor, which I hope the authors may consider in their revision.

Line 39 – The definition of the “Competitive exclusion principle” might be updated here, in particular to specify that an important tenet of the principle is that competitors have the same/similar resource requirements. That is, a community without direct stabilizing interactions may support multiple concurrent species so long as their nutrition/resource requirements are sufficiently dissimilar. I note that the caption of Figure 1A subsequently defines the competitive exclusion principle with this requirement (i.e. competition for common resources). If this change is made perhaps line 41 could be toned down somewhat, e.g. that it can be “difficult” to maintain community composition (rather than impossible).

We have modified the definition of the competitive exclusion principle in the main text and toned down the statement on the difficulty of maintaining community composition.

Line 50 – Somewhere in this paragraph references to DOIs 10.1002/biot.202100169 and 10.1038/s41565-021-00878-4 could be added. These studies demonstrate some attempts at computational control of a cellular communities (admittedly not realised as elegantly with optogenetics, and with a much more limited scope and control/modelling technique compared to the present study).

We thank the reviewer for pointing out these missing references. We have now modified the paragraph (splitting it into two paragraphs and reformulating some sentences) to accommodate both of these citations.

Line 171 – The detailed description of the evotron setup is much appreciated. Do you have any sense of how it compares to the recently published ReacSight toolbox (DOI 10.1038/s41467-022-31033-9)? A detailed discussion of this comparison is likely not essential to the paper's core message, however perhaps readers would appreciate a reference to this background/alternative should they aim to implement a similar approach in their own work.

We would like to thank the reviewer for pointing this out. We have now added a few lines about ReacSight framework at the end of the evotron section in the main text, and few more details at the end of Supplementary Section 2.

Line 307 – typo "... same number of free parameters AS their conventional..."

We have corrected the typo.

Line 313 – Is there a reason this figure is referenced as "A (Left)..." etc rather than A,B,C,D...? I will leave it to the authors' final judgement, but perhaps it is easier and faster for the reader to interpret if each sub-panel is conventionally labelled with a letter.

We can see how multiple sub-panel labeling could facilitate the interpretation of the caption and thank the reviewer for their suggestion. However, we would prefer to keep the three plots as a single sub-panel, because we believe this makes it visually more clear that the color legend and equation shown in sub-panel A apply to all three plots.

Line 365 – Axis labels on the small graphs in Panel B were small/difficult to read when printed. Font size could be increased here?

We thank the reviewers for pointing this out. We have increased the font size of the axis labels to improve readability.

Line 405 – Apologies if this stated somewhere earlier that I missed; why is the evotron framework only able to control five independent vials in this case? My understanding was the eVolver platform had capability for sixteen vials in parallel. Would it be possible to comment on the limiting factors in the evotron approach's throughput (is it due to time required for each sample and subsequent cleaning?). Perhaps this could be discussed in Supplementary Section 2 where evotron is described in more detail.

This is an important detail related to our setup operation - thanks for bringing this point to our attention. This is indeed the case that the throughput in our co-culture composition control experiments are limited by the control input update frequency (once

every 30 minutes) and the time taken for each automated sampling, measurement and cleaning process (approx. 5 and half minutes). We have now added this information as a note at the end of Supplementary Section 2.

Line 408 / Line 465 – Here (and elsewhere) the importance of the a priori host-aware modelling is stressed. E.g. on line 465 that it was “responsible for the excellent performance of our closed-loop control. Furthermore, it considerably reduced the final experimental effort,”. I agree completely with the authors’ assessment that this host-aware framework seems effective and easy-to-implement. However, it is difficult to justify that this was the one factor responsible for the excellent performance of the closed-loop control. For example, if you had taken some open-loop step responses (e.g. as in Fig 4D) and fit a simple one-ODE system to this, could this have been used to computationally tune a PID with (roughly) similar outcome (and even less effort)? This question seems important for the target audience of the paper (e.g. those working in control application fields as outlined in the Discussion of the manuscript), who may wish for an as-simple-as-possible methodology that achieves reasonable outcome. Perhaps the authors could comment on the potential pitfalls of using a simpler modelling framework in PID tuning – in what situations might it be (or not be) good enough? Perhaps this could include some commentary on how such modelling approaches might perform should a MPC approach be used rather than PID (as done in some of the authors’ past work on cybergenetics control).

We thank the reviewer for this very valid point. We have added a discussion on the possibility of using simpler models for optimization of controller parameters, as well as MPC.

Congratulations again to the authors on their excellent work!

Harrison Steel

Reviewer #2 (Remarks to the Author):

This manuscript details work that allows one to externally control the ratio of two bacterial strains growing in the same turbidostat. To do this, the authors use a PID controller to measure and adjust the growth rate of one of the two cell types. The growth rate is adjusted using a light-sensitive split T7 RNAP that regulates the expression of chloramphenicol acetyltransferase, an enzyme that confers resistance to chloramphenicol. Overall, I think this is pretty stellar work and should be of interest to many in the fields of synthetic biology and metabolic engineering.

I usually would now list all the faults I find with the paper, but, in truth, I didn't really find anything glaring. The experiments are done well and the appropriate controls look solid. The control theory is also very well explained and should be reproducible. I might give it a try in my lab if I can get my people to tinker with our turbidostats.

Also, when I first read the paper, I had thought Mario Di Bernardo had done almost exactly this using microfluidics. But looking back, I can't find it, just the theory paper his group did in ACS Syn Bio.

Nicely done!

We thank this reviewer deeply for his positive appreciation and kind support!

Matthew R. Bennett
Rice University